# Making the Rounds: Exploring the Role of Circulating Tumor DNA (ctDNA) in Non-Small Cell Lung Cancer

**DOI:** 10.3390/ijms23169006

**Published:** 2022-08-12

**Authors:** Misty Dawn Shields, Kevin Chen, Giselle Dutcher, Ishika Patel, Bruna Pellini

**Affiliations:** 1Department of Internal Medicine, Division of Hematology/Oncology, Indiana University School of Medicine, Indiana University Melvin and Bren Simon Cancer Center, Indianapolis, IN 46202, USA; 2Department of Radiation Oncology, Division of Cancer Biology, Washington University School of Medicine, St. Louis, MO 63110, USA; 3Department of Medicine, Division of Solid Tumor Oncology, University Hospitals Seidman Cancer Center, Case Western Reserve University, Cleveland, OH 44106, USA; 4Department of Public Health, University of South Florida, 12902 Magnolia Drive, Tampa, FL 33612, USA; 5Department of Thoracic Oncology, Moffitt Cancer Center and Research Institute, Tampa, FL 33612, USA; 6Department of Oncologic Sciences, Morsani College of Medicine, University of South Florida, Tampa, FL 33612, USA

**Keywords:** circulating tumor DNA, cell-free DNA, liquid biopsy, molecular profiling, treatment response monitoring, minimal residual disease, MRD, early cancer detection, non-small cell lung cancer, NSCLC

## Abstract

Advancements in the clinical practice of non-small cell lung cancer (NSCLC) are shifting treatment paradigms towards increasingly personalized approaches. Liquid biopsies using various circulating analytes provide minimally invasive methods of sampling the molecular content within tumor cells. Plasma-derived circulating tumor DNA (ctDNA), the tumor-derived component of cell-free DNA (cfDNA), is the most extensively studied analyte and has a growing list of applications in the clinical management of NSCLC. As an alternative to tumor genotyping, the assessment of oncogenic driver alterations by ctDNA has become an accepted companion diagnostic via both single-gene polymerase chain reactions (PCR) and next-generation sequencing (NGS) for advanced NSCLC. ctDNA technologies have also shown the ability to detect the emerging mechanisms of acquired resistance that evolve after targeted therapy. Furthermore, the detection of minimal residual disease (MRD) by ctDNA for patients with NSCLC after curative-intent treatment may serve as a prognostic and potentially predictive biomarker for recurrence and response to therapy, respectively. Finally, ctDNA analysis via mutational, methylation, and/or fragmentation multi-omic profiling offers the potential for improving early lung cancer detection. In this review, we discuss the role of ctDNA in each of these capacities, namely, for molecular profiling, treatment response monitoring, MRD detection, and early cancer detection of NSCLC.

## 1. Introduction

Revolutionary developments in ultrasensitive sequencing approaches have led to the successful implementation of liquid biopsies to inform clinical decisions in patients with advanced NSCLC [1,2,3,4]. Liquid biopsies provide a minimally invasive method of cancer assessment via the analysis of circulating analytes, such as circulating nucleic acids (circulating tumor DNA (ctDNA) and cell-free RNA (cfRNA)), circulating tumor cells (CTCs), microRNAs, extracellular vesicles, tumor-educated platelets, and tumor-specific cell-free DNA methylation. These analytes can be found in several biological fluids, including plasma, serum, urine, pleural fluid, saliva, and cerebrospinal fluid (CSF) [3,5,6,7,8,9,10,11]. The presence of cell-free nucleic acid fragments in the blood was first described by Mandel and Metais in 1948 [12]. Cell-free DNA (cfDNA) consists of fragments of nonencapsulated extracellular DNA, which usually measure between 150 and 200 bp in length and are passively released into the blood by apoptotic and necrotic cells [13,14,15]. Plasma cfDNA is mostly released by endothelial and peripheral blood mononuclear cells (PBMCs) [14,15,16,17]; in patients with cancer, a small fraction of cfDNA consists of DNA fragments released into the blood by tumor cells, termed ctDNA [18]. Although several studies have demonstrated that serum has higher concentrations of cfDNA compared to plasma, most of the cfDNA in serum originates from lysed PBMCs, which makes it more challenging to isolate the ctDNA [17,19,20,21]. Therefore, in an effort to reduce PBMC contamination, plasma samples are preferred for ctDNA analysis [17]. While plasma ctDNA is relatively prevalent in most patients with advanced solid malignancies (except CNS tumors), the sensitive detection of ctDNA has been historically challenging, largely due to the short half-life of ctDNA (15 min to 2.5 h), and assay specifications are required to mitigate the rate of false-positive variant-calling from clonal hematopoiesis (CH) and background sequencing error [7,22,23]. With the development of next-generation sequencing (NGS), minimally invasive genotyping via ctDNA now plays an increasingly essential role in clinical practice, particularly via molecular profiling, to inform treatment decisions and serves as a powerful molecular response metric for translational research and precision medicine [3,24]. In fact, two commercially available NGS-based assays (Guardant360^®^, Guardant Health, Redwood City, CA, USA and FoundationOne^®^ Liquid CDx, Foundation Medicine INC., Cambridge, MA, USA) have received approval in 2020 by the US Food & Drug Administration (FDA) for identifying genomic alterations in the plasma of patients with advanced-stage solid malignancies, including NSCLC [25,26]. Beyond molecular profiling, minimally invasive genotyping via ctDNA for NSCLC is under active investigation for treatment-response monitoring, minimal residual disease (MRD) detection, and early cancer detection [13,15]. In this review, we outline the current applications of ctDNA in the management of NSCLC and discuss future directions for improving personalized care (Figure 1).

## 2. Pre-Analytical Conditions for ctDNA Assessment

ctDNA is a highly sensitive, minimally invasive resource for the qualitative and quantitative molecular profiling of NSCLC. However, the assessment of ctDNA introduces substantial challenges and drawbacks, including modifiable (e.g., technical considerations) and non-modifiable variables (e.g., tumor heterogeneity) that impact the sensitivity of its analysis (Table 1). Here, we aim to highlight key pre-analytical conditions that influence the performance of ctDNA assays, including the integral aspects of venipuncture, collection tubes and preservatives for cellular stability, and specimen processing time for successful analysis.

Although no standard collection volume has been established for ctDNA analysis, given the low concentration of ctDNA in plasma (concentration, 1 to 18 ng per mL), the collection of large volumes of blood (typically, 10 to 18 mL) are recommended to increase analytical sensitivity [27,28]. Phlebotomies are to be performed via slow extraction from the patient to prevent the hemolysis of surrounding leukocytes [29]. Samples can be directly collected in K2EDTA tubes (i.e., BD Vacutainer^®^ Blood Collection Tubes) if there is a limited shelf-life for storage (e.g., less than 6 h) prior to the isolation of ctDNA. However, with K2EDTA collection, if plasma preparation by centrifugation is delayed (e.g., by more than 24 h from venipuncture), noticeable DNA contamination will be apparent within the sample from presumed leukocyte lysis, resulting in false elevations in cfDNA content and poor sample quality [30]. Alternatively, newer formulations of tubes (e.g., Streck cfDNA BCT^®^ and PAXgene Blood ccfDNA Tubes) containing proprietary preservatives for cellular integrity are permissible for shipping at ambient temperature, with ctDNA stability of up to one week from venipuncture [30,31].

Timing, temperature, and sample collection represent a snapshot of important pre-analytical conditions that can significantly impact the quality of ctDNA analysis. As ultra-sensitive techniques become a mainstay of clinical practice for ctDNA analysis, rigorous standardized methods for collection and processing with either guideline-based or institutional CLIA-based protocols are highly recommended.
ijms-23-09006-t001_Table 1Table 1**ctDNA** **detection techniques.**TechnologyExamplesAssay PersonalizationReported LOD(% of cfDNA)Potential Clinical UsesLimitationsAS-PCRARMS [32],Cobas EGFR mutation test v2 (Roche) [33]Some Required~0.1–1Molecular ProfilingDetects only known mutations or a small number of variants concurrently; low sensitivitydPCRdPCR [34],ddPCR [35],BEAMing [36]Some Required~0.01Molecular Profiling, Treatment Monitoring, ctDNA MRDDetects only known mutations or a small number of variantsMultiplex PCR-based NGSTAm-Seq [4], TAm-Seq Enhanced [37],Safe-SeqS [38],Natera^®^ [39]Some Required~0.01–2.0Molecular Profiling, Treatment Monitoring, ctDNA MRDDetects only known mutations; less comprehensive than other NGS methods; incapable of detecting SCNAs and fusions without assay personalizationHybrid captured-based NGSCAPP-Seq [40,41],TEC-Seq [42], Guardant360^®^ [25,43,44], FoundationOne^®^ Liquid CDx [26,45]Not Required~0.001–0.5Molecular Profiling, Treatment Monitoring, ctDNA MRDLess comprehensive than WGS and WESWESWES [46]Not Required~5Molecular Profiling, ctDNA MRDLow sensitivity; high costWGSWGS [47,48]Not Required~10Molecular Profiling, ctDNA MRDLow sensitivity; mostly limited to SCNA detection; high costcfDNA methylationcfMeDIP-seq [49],cfMBD-seq [50],GRAIL Galleri™ [51]Not RequiredVariableEarly Lung Cancer Detection, ctDNA MRDVariable/limited sensitivity for early-stage disease detectionFragmentomicsDELFI [52]Not RequiredVariableEarly Lung Cancer Detection, ctDNA MRDVariable/limited sensitivity for early-stage disease detectionCombined approachesCAPP-Seq + GRP [53], CancerSEEK [54]VariableVariableEarly Lung Cancer Detection, ctDNA MRDMay require more time and resourcesAbbreviations: AS-PCR, allele-specific PCR; AMRS, Amplification Refractory Mutation System; dPCR, digital PCR; ddPCR, digital droplet PCR; BEAMing, beads, emulsion, amplification, and magnetics; ctDNA, circulating tumor DNA; MRD, minimal residual disease; NGS, next-generation sequencing; TAm-Seq, Tagged-Amplicon sequencing; Safe-SeqS, Safe Sequencing System; SCNAs, somatic copy-number alteration; CAPP-Seq, Cancer Personalized Profiling by Deep Sequencing; TEC-Seq, Targeted Error Correction Sequencing; GRP, Genome Representation Profiling; WES, whole exome sequencing; WGS, Whole Genome Sequencing; cfMeDIP-seq, Cell-Free methylated DNA immunoprecipitation-sequencing; cfMBD-seq, cell-free methyl-CpG binding proteins; DELFI, DNA evaluation for early interception; cfDNA, cell-free DNA. *Adapted from* [23] Thoracic Surgery Clinics, Pellini, B.; Szymanski, J.; Chin, R.-I.; Jones, P.A.; Chaudhuri, A.A. Liquid Biopsies Using Circulating Tumor DNA in Non-Small Cell Lung Cancer. *Thorac. Surg. Clin.* **2020**, *30*, 165–177. Copyright 2020, with permission from Elsevier, and adapted with permission from Springer Nature [55]: *Springer Molecular Diagnostics & Therapy*. Chin, R.-I.; Chen, K.; Usmani, A.; Chua, C.; Harris, P.K.; Binkley, M.S.; Azad, T.D.; Dudley, J.C.; Chaudhuri, A.A. Detection of Solid Tumor Molecular Residual Disease (MRD) Using Circulating Tumor DNA (ctDNA). *Mol. Diagn. Ther.* **2019**, *23*, 311–331. Copyright 2019.

## 3. Molecular Profiling

Over the past two decades, pivotal discoveries in lung cancer biology and landmark clinical trials with the introduction of targeted therapy and immunotherapy have led to major transformations in the treatment paradigm for NSCLC [56,57]. Historically, patients with NSCLC were treated with first-line histology-based cytotoxic doublet chemotherapies [58,59]. With the discovery of specific activating gene alterations (e.g., *EGFR*), preclinical development and the clinical validation of respective targetable therapeutics for oncogene-driven NSCLC has led to the approval of new targeted therapies [56,60,61,62,63,64,65]. As the focus of NSCLC therapy shifted to a more personalized approach, knowledge of molecular biomarker status has essentially become a prerequisite for providing comprehensive care in thoracic oncology [66,67]. Traditionally, the detection of oncogenic driver alterations has required available and adequate tumor tissue to perform immunohistochemistry (IHC) or single-gene polymerase chain reaction (PCR) assays [32,56,66]. Single-gene quantitative PCR (qPCR)-based assays can be time-consuming and exhaustive to test on tissue across the vast array of actionable mutations (Table 1). Moreover, tissue-based genotyping after disease progression while on therapy requires obtaining a tissue re-biopsy that could be challenging for various reasons (e.g., technically difficult tumor locations) [23]. Furthermore, qPCR-based assays are inherently confirmatory for a genotype-informed allele-specific alteration (e.g., *EGFR* L858R) but are not investigative for rare sequence alterations [68].

A major advancement in the field of oncology was the development of minimally invasive methods of tumor genotyping, including digital droplet PCR (ddPCR) and next-generation sequencing (NGS). ddPCR is a digital PCR technology that emulsifies plasma DNA into thousands of droplets, which are then amplified using PCR methods, fluorescently labeled, and read with an automated droplet flow cytometer [34,69,70]. Oxnard et al. demonstrated via proof-of-principle the utility of ddPCR to perform minimally invasive genotyping of plasma ctDNA samples from patients with *KRAS*-mutant and *EGFR*-mutant NSCLC, showcasing the high sensitivity and specificity of this technology [71]. In contrast to ddPCR, targeted NGS utilizes a gene panel to query multiple plasma ctDNA mutations in parallel [23,55]. While these assays generally have high sensitivity and specificity for detecting single nucleotide variants (SNVs), initial NGS-based methods had insufficient sensitivity for the detection of chromosomal copy-number alterations and rearrangements (Table 1) [46,72]. To improve sensitivity and reduce error rates, massive parallel sequencing techniques, such as the Safe-Sequencing System (Safe-SeqS) [38] and error-suppressed multiplexed deep sequencing [37,73], were developed (Table 1). In a study of 48 patients with advanced NSCLC, bias-corrected targeted NGS interrogating 11 targetable oncogenic driver genes in NSCLC was applied [74]. Mutations were identified to a depth of ≥0.4% of variant allele frequency (VAF), with 100% sensitivity and specificity [74]. In addition, this NGS approach identified a novel double-deletion in the *EGFR* exon 19 and *MET* amplification that were previously missed by direct tissue sequencing despite four biopsies [74].

To overcome limitations in terms of cost, reduce the need for patient-specific optimization, and improve the sensitivity of detection (Table 1), Newman et al. developed CAPP-Seq (Cancer Personalized Profiling by deep Sequencing), a hybrid capture-based method of targeted NGS for ctDNA detection [40]. CAPP-Seq can detect all major classes of genomic alterations, including SNVs, indels, copy-number variations (CNV), and rearrangements, with levels of detection for mutant allele fractions down to ~0.02% [40,41,68]. CAPP-Seq ctDNA profiling was employed to study resistance mechanisms in 43 patients with advanced NSCLC who were treated with rociletinib, a third-generation EGFR TKI [75]. Using a biotinylated probe (selector) targeting 252 genes, the authors identified multiple putative resistance mechanisms that emerged during rociletinib treatment in 46% of patients, including a novel *EGFR* L798I mutation [75]. These findings showcase the potential of targeted NGS to detect mechanisms of resistance.

Historically, molecular rearrangements (i.e., fusions) are detected by utilizing either fluorescence in-situ hybridization (FISH) or IHC from tumor tissue specimens. However, due to operator dependency, the heterogeneity of breakpoints, small intrachromosomal deletion events, and increasing numbers of novel fusion partner genes, clinical detection of molecular rearrangements by these methods carries high rates of false-positive and false-negative results [76,77]. Anchored multiplex PCR (AMP), a NGS target enrichment method, was developed by Zheng et al. for rapid and efficient gene arrangement detection [78]. This technique was applied to nucleic acids extracted from 319 formalin-fixed paraffin-embedded (FFPE) samples from different solid tumors, including lung adenocarcinoma, and demonstrated 100% sensitivity and specificity to detect gene rearrangements when compared with standard clinical FISH assays [78]. Moreover, the authors demonstrated its potential for identifying new therapeutically important gene fusions in NSCLC [78]. In another study, tumor-specific alterations in *ROS1* were identified by proprietary Guardant360^®^ NGS technology, utilizing plasma from patients with *ROS1*-positive NSCLC [79]. The assay demonstrated 100% concordance in the detection of *ROS1* fusions between tissue and plasma genotyping performed for seven patients [79]. CD74 predominated as the most common fusion partner, among seven novel fusion partners that were identified [79].

To understand if targetable kinase fusions can be reliably identified in plasma with comprehensive genomic profiling, the pan-tumor landscape of 16 targetable kinase fusions was tested on and validated across 36,916 blood samples and 368,931 tumor tissue samples, utilizing proprietary FoundationOne^®^ NGS technology [45]. The investigators reported that 88% of patients had detectable ctDNA in plasma, among whom 1.8% had detectable kinase fusions [45]. A 92% concordance rate was noted in the 63 available paired tissue and plasma samples, validating the utility of NGS technologies to detect ctDNA-derived fusions [45]. In fact, among 32 patients with fusions detected in plasma but not in tissue, 66% had new fusions that represented mechanisms of acquired resistance to therapy [45].

Commercially available multigene panels have revolutionized patient care in the 21st century, leading to the FDA approval of two liquid biopsy assays (Table 1): Guardant360^®^ CDx [25,43,44] and FoundationOne^®^ Liquid CDx [26,80]. In 2020, both of these hybrid capture-based, targeted NGS assays were approved for clinical application to identify genomic alterations in patients with advanced-stage solid malignancies. To address the suboptimal detection of ctDNA in patients with NSCLC who were afflicted with brain metastases, a prospective study explored the utility of radiographic metrics and NGS testing on CSF for ctDNA detection [81]. Further clinical development of predictive modeling is needed to better understand the features of brain metastases, which may improve the detection of CSF ctDNA [81].

## 4. Treatment Response Monitoring

The introduction of targeted therapy for patients with advanced NSCLC has prompted an ongoing search to identify personalized biomarkers that predict which patients will have durable clinical benefits with treatment [82,83]. Unfortunately, selective pressure with targeted therapy drives molecular evolution, wherein the monitoring of response and the detection of mechanisms of acquired resistance are necessary for the expedient management of patients after disease progression while on therapy [84,85].

A study by Murtaza et al. enrolled six patients with advanced breast cancer, ovarian cancer, and NSCLC who were serially followed over 1 to 2 years, using the exome-wide analysis of plasma ctDNA to quantify mutation allele fractions [46]. Patient LUN-209, with advanced NSCLC, progressed on treatment with gefitinib after 109 days, corresponding with the emergence of a resistance-conferring *EGFR* T790M mutation detected by digital PCR of plasma ctDNA. This result established proof-of-principle that plasma ctDNA can be used to identify emergent mutations involved in the clonal evolution of acquired drug resistance in NSCLC [46].

FASTACT-2 [86], a randomized study of intercalated erlotinib versus placebo use among patients receiving 6 cycles of platinum-based therapy for advanced NSCLC, included serial plasma collection from 305 patients at baseline, after cycle 3, and at disease progression to assess for *EGFR* mutations using the cobas^®^ EGFR Mutation Test, a real-time PCR test [33,86,87]. Blood-based testing for *EGFR* ctDNA exhibited 75% sensitivity and 96% specificity, as well as 88% concordance to matched tumor tissue [87]. A median progression-free survival (PFS) benefit was observed in patients with detectable *EGFR* mutations in ctDNA vs. *EGFR* wild-type ctDNA at baseline who had received erlotinib (13.1 vs. 6.0 months; HR, 0.22; 95% CI: 0.14–0.33) [87]. Interestingly, patients with detectable *EGFR* mutations in blood plasma at baseline and who did not clear the *EGFR* mutation after 3 cycles had a shorter PFS (7.2 vs. 12.0 months; HR, 0.32; 95% CI: 0.21–0.48) and worse overall survival (OS) (18.2 vs. 31.9 months; HR, 0.51; 95% CI: 0.31–0.84) [87].

A retrospective study performed by Phallen et al. assessed serial liquid biopsies from 28 patients with advanced NSCLC, who were undergoing treatment with targeted TKIs, to measure the ctDNA dynamics and predict clinical outcomes [88]. Specifically, blood draws performed at baseline and serial timepoints were analyzed with ultrasensitive targeted error correction sequencing (TEC-Seq) and whole-genome sequencing to identify tumor-derived alterations in ctDNA, such as somatic variants and chromosomal copy-number alterations (Table 1) [88]. The results revealed a bimodal distribution of cell-free tumor load after the initiation of TKI therapy (median time of 19 days), whereas ctDNA responders showed the near-complete elimination of cell-free tumor load at early follow-up (cell-free tumor load reduction > 98%) compared to ctDNA non-responders (cell-free tumor load reduction ≤ 98%). The ctDNA responders had a significantly prolonged PFS (median of 13.7 vs. 1.6 months; HR 66.6; 95% CI: 13.0–341.7 months) [88]. Tumor response, including ctDNA non-responders, was detected on average 4 weeks earlier than radiographic assessment, highlighting the potential role of comprehensive genome-wide ctDNA analysis for the earlier evaluation of tumor burden in response to targeted therapies [42,88,89].

LungBEAM, a prospective study across 19 Spanish hospitals, collected serial blood specimens at baseline from 110 treatment-naïve patients with advanced *EGFR*-mutant NSCLC and monthly for *EGFR* T790M emergence while on first-line EGFR TKI therapy until radiographic progression [90]. Progression was observed in 68 of the 110 patients (61.8%) who completed the study. Twenty-six patients (23.6%) showed the presence of an *EGFR* T790M mutation in plasma, with detection at a median of 76 days prior to radiographic progression [90].

The ALEX study (NCT02075840) compared two ALK inhibitors, alectinib and crizotinib, in the first-line setting via a randomized trial for patients with advanced ALK-positive NSCLC [91]. A retrospective analysis utilizing samples from the ALEX study found a positive correlation between cfDNA concentration and the number of lesions, organ lesion sites, and tumor size. OS data remains immature [92]. Further prospective trials utilizing ctDNA to predict treatment responses in NSCLC are underway (Table 2).

The search for the perfect biomarker to predict disease response to immune checkpoint inhibition (ICI) and demonstrate improvements in OS remains elusive [93,94,95]. Several novel candidate biomarkers have been investigated, including tumor PD-L1 [93,96], tumor mutational burden (TMB) [97,98], mismatch repair deficiency [99], mutations in *FAT1* [100], and lymphocyte infiltration in the tumor microenvironment [94,101], with mixed evidence to accurately predict the response to ICI [95]. A proof-of-concept prospective study (NCT02866149) in patients with advanced NSCLC, uveal melanoma, or microsatellite-unstable colorectal cancer (CRC), who were treated with standard-of-care anti-PD-1 therapy (pembrolizumab or nivolumab) compared ctDNA monitoring at baseline and week 8 of treatment, with tumor response established by radiological evaluation [102]. The ctDNA levels were detected via bidirectional pyrophosphorolysis-activated polymerization (bi-PAP) assay, ddPCR, or targeted-NGS with a focused panel of 39 cancer-related genes. At baseline, ctDNA was detected in 10 of 15 patients [102]. ctDNA detection at week 8 of treatment was a significant prognostic factor for shorter PFS (HR, 10.2; 95% CI: 2.5–41) and OS (HR, 15; 95% CI: 2.5–94.9). This landmark study demonstrated the role of ctDNA monitoring as a tool for treatment monitoring with anti-PD-1 immune checkpoint inhibitors [102].

One challenge that exists in the field of immuno-oncology is the interpretation of early radiographic response and misinterpretation of imaging, given potential pseudo-progression or delayed tumor shrinkage with initial cycles of treatment [103,104]. To offer earlier indications of efficacy with ICI, a single institution study investigated serial ctDNA levels by high-throughput DNA sequencing in 28 patients diagnosed with metastatic NSCLC, who were actively receiving immunotherapy (anti-PD-1 or anti-PD-L1 alone, or with other immunotherapy agents) [105]. CT scans were obtained within 30 days before the first treatment and serially at 6- to 12-week intervals, or when clinically indicated. Blood samples were collected on day 1 (prior to treatment) and with routine blood draws, typically at intervals of 2 or more weeks [105]. Concordance between ctDNA response and radiographic response was observed (Cohen’s kappa, 0.753) [105]. ctDNA response (>50% reduction in level from baseline) was observed earlier than the radiographic response per RECIST version 1.1 (median of 24.5 vs. 72.5 days), where the time undergoing treatment was significantly prolonged in ctDNA responders versus non-responders (median of 205.5 vs. 69 days, *p* < 0.001) [105]. While on ICI, ctDNA response was associated with improved PFS (HR, 0.29; 95% CI: 0.09–0.89) and superior OS (HR, 0.17; 95% CI: 0.05–0.62) [105].

In 2018, Raja et al. demonstrated that changes in the VAFs of somatic mutations in ctDNA were strongly correlated to the outcomes of patients treated with durvalumab [106]. Samples from 28 patients were utilized from Study 1108 (NCT01693562), which investigated the role of durvalumab in patients with advanced NSCLC who had progressed, been ineligible for, or refused any number of prior therapies [106]. Independent validation was performed in 72 patients with advanced *EGFR* wild-type NSCLC from cohort 2 of the ATLANTIC study (NCT02087423), which assessed the effect of durvalumab for the treatment of patients with Stage IIIB/IV NSCLC, who progressed following two or more systemic treatments [107]. One or more somatic variants, including SNV, indels, fusions, and silent mutations, were detected in the ctDNA of 27 of 28 patients (96%) at pre-treatment baseline [106]. Patients with a decrease in mean VAF had a median treatment duration of 22 months compared to 6.5 months for patients with an increase in mean VAF (*p* < 0.001) [106].

Among three phase-I/II studies (Study 1108, ATLANTIC, and Study 10) with durvalumab alone or in combination with tremelimumab, conducted across 16 advanced-stage tumor types, Zhang et al. demonstrated similar findings where pre-treatment and on-treatment ctDNA dynamics serve as prognostic and predictive biomarkers for long-term survival on ICI, respectively, coining the concept of a “molecular response metric”, using ctDNA as a valuable tool to complement radiographic changes in trial endpoints [108]. Similarly, Anagnostou et al. demonstrated that among 24 patients with metastatic NSCLC treated with ICI, ctDNA non-responders had shorter PFS (median of 5.2 vs. 14.5 months; HR, 5.36; 95% CI: 1.57–18.35) and OS (median of 8.4 vs.18.7 months; HR, 6.91; 95% CI: 1.37–34.97) compared to ctDNA responders [109]. Moreover, ctDNA response detected disease progression on average 8.7 weeks earlier than CT imaging [109].

In another study by Ricciuti et al., where 62 patients with advanced NSCLC were treated with first-line pembrolizumab, plus or minus platinum-based chemotherapy, significant correlations between rapid decreases in ctDNA with higher response rates (60.7% vs. 5.8%, *p* = 0.0003), improvements in median PFS (8.3 vs. 3.4 months, HR, 0.29; 95% CI: 0.14–0.60, *p* = 0.0007), and median OS (26.2 vs. 13.2 months, HR, 0.34; 95% CI: 0.15–0.75, *p* = 0.008) were noted [98]. Comparable results were reported in a single-center, prospective study at the University of Pennsylvania investigating the role of serial ctDNA levels as a predictive biomarker for therapeutic response to pembrolizumab-based therapy (i.e., monotherapy or in combination with chemotherapy) [110]. Specifically, 67 patients had plasma samples collected at baseline (prior to treatment initiation) and at 9 weeks of treatment. The samples were analyzed via 74-gene panel NGS assays (Guardant360^®^ and GuardantOMNI™) [110]. Patients who achieved a molecular response, defined by >50% decrease in mean VAF, showed meaningful durable clinical benefit (DCB) with significantly longer PFS (HR, 0.25; 95% CI: 0.13–0.50) and OS (HR, 0.27; 95% CI: 0.12–0.64), compared to molecular non-responders [110].

In an effort to refine ctDNA prediction modeling with a novel multiparameter biomarker for long-lasting immunotherapy response, Nabet et al. developed and validated a robust minimally invasive multiparameter assay to reflect the immune milieu, termed DIREct-On (durable immunotherapy response estimation by immune profiling and ctDNA-On treatment) [111]. Pre-treatment specimens were collected from 99 patients with advanced NSCLC, who were treated with anti-PD-L1, including tumor biopsies for PD-L1 expression, plasma ctDNA for analysis using CAPP-Seq, and leukocyte profiling by CIBERSORT and/or flow cytometry [111]. Patients were split into 3 cohorts for early on-treatment downstream analysis (within 4 weeks from start, median 2.4 weeks), including (1) feature discovery analysis (*n* = 22), (2) training set for DIREct Discovery Cohort (*n* = 34), or (3) testing set for DIREct Validation Cohort (*n* = 38) [111]. Of 99 patients, 94 had detectable pre-treatment ctDNA with CAPP-Seq [111]. Investigators demonstrated that lower pre-treatment ctDNA levels and lower baseline circulating CD8 T cell levels (accuracy = 70%) are independent variables associated with DCB [111]. Early on-treatment ctDNA levels decreased > 50% from baseline in 59% of patients, and this metric outperformed all individual pre-treatment factors to predict DCB (*p* < 0.05, accuracy = 73%). The ctDNA dynamics were also significantly associated with improved PFS (*p* = 0.013, HR = 2.28) [111]. Utilizing a Bayesian approach incorporating tumor-cell intrinsic and extrinsic determinants for ICI benefits, including PD-L1 expression, normalized bTMB, circulating CD8 T cells for DIREct-Pre, plus ctDNA dynamics for DIREct-On were trained, tested, and validated [111]. Investigators reported that patients with higher DIREct-On scores had significantly longer PFS than those with low DIREct-On scores (mPFS 8.1 vs. 2.1 months, respectively, *p* < 0.0001, HR = 7.11) [111]. DIREct-On demonstrated superiority in the prediction of PFS, compared to ctDNA dynamics (*p* < 0.01), CD8 T cell fraction (*p* < 0.01), and bTMB (*p* < 0.05) alone. Prospective trials utilizing DIREct-On to validate clinical utility are anticipated [111].

## 5. Minimal Residual Disease (MRD) Detection

MRD is defined as a microscopic volume of tumor cells that remain after curative-intent treatment, yielding significantly lower levels of ctDNA in plasma (often below 0.1% of total cfDNA) relative to advanced-stage disease [55]. In the MRD setting, either tumor-informed or tumor-naïve assays can be used to detect ctDNA [112,113]. Tumor-informed assays utilize prior mutational knowledge via the sequencing of tumor tissue to interrogate plasma samples for patient-specific mutations [39,114,115]. Tumor-naïve assays perform *de novo* variant-calling using a preselected panel of hotspot mutations to query multiple loci simultaneously across a genomic scale [116,117]. A tumor-informed approach is associated with certain advantages (e.g., customization to an individual patient and ability to track multiple simultaneous tumor-specific mutations) and disadvantages (e.g., limited clonal variant detection and increased turnaround time) compared to tumor-naïve assays. Nonetheless, both approaches, in conjunction with target enrichment (e.g., PCR or hybrid-capture) and false-positive error suppression (e.g., sequencing of PBMCs to filter CH, molecular barcoding, and in silico elimination of stereotypical artifacts), seek to minimize background noise and enable greater sequencing depths of tumor-derived mutations for highly sensitive MRD detection (Table 1) [112,113]. Although commercially available assays for MRD detection have not yet been approved by the FDA to be used in clinical practice in patients with NSCLC, several groups have sought to demonstrate ctDNA MRD as a viable biomarker for predicting disease relapse and treatment response, particularly for patients with NSCLC after curative-intent treatment (Figure 2) [39,112,116,117].

In 2017, Abbosh et al. demonstrated that ctDNA MRD detection after surgical resection in patients with stages I-III NSCLC was associated with disease relapse. The twenty-four patients included in this analysis were enrolled onto the TRACERx study (NCT01888601), a multicenter prospective trial designed to study the evolutionary dynamics of NSCLC in over 800 patients [39]. The authors applied Signatera™ (Natera, San Carlos, CA, USA), a tumor-informed assay that utilizes a patient-specific multiplex PCR to sequence cfDNA, to the plasma of 24 patients with stage IA-IIIB NSCLC, following surgical resection plus or minus adjuvant therapy and post-operative radiation [39]. Among 14 patients with disease relapse, five patients (36%) had detectable ctDNA at the first timepoint after surgery (i.e., within 30 days) [39]. Upon longitudinal plasma sampling from these patients (i.e., surveillance), 93% of the patients who eventually developed disease recurrence had detectable ctDNA MRD before or at the time of clinical relapse [39]. For the 10 patients who remained relapse-free, ctDNA MRD was not detected in 90% and 70% of patients at the first timepoint of collection and in the surveillance setting, respectively [39]. Of note, the single patient with a false-positive detection of ctDNA at the first timepoint after surgery then received both adjuvant chemotherapy and radiation therapy, which likely ablated MRD since ctDNA became undetectable at three serial timepoints following the completion of adjuvant therapy [39]. Conversely, three patients with detectable ctDNA at the first timepoint after surgery had rising ctDNA levels following adjuvant chemotherapy, and all three patients relapsed within one year [39]. Altogether, these findings show that ctDNA MRD, as well as ctDNA dynamics, may inform clinicians regarding the risk of disease relapse following the curative-intent treatment of resectable NSCLC.

Chaudhuri et al. published the first study demonstrating the promise of ctDNA MRD as a prognostic biomarker for patients with unresectable NSCLC who underwent curative-intent treatment [116]. The authors applied CAPP-Seq, which utilizes hybrid capture-based NGS with a panel of 128 genes that are recurrently mutated in lung cancer, to sequence cfDNA from the plasma of 37 patients with stage IB-IIIB NSCLC, most of whom underwent concurrent chemoradiation (CRT), alongside three patients with limited-stage small cell lung cancer [116]. Among 32 patients with evaluable samples, 17 patients had detectable ctDNA at the first timepoint within four months of treatment completion [116]. These patients had a significantly shorter freedom from progression (FFP) and disease-specific survival (DSS) rate relative to those with undetectable ctDNA at the same timepoint [116]. Moreover, all 14 patients who remained relapse-free had undetectable ctDNA, showcasing the high specificity of CAPP-Seq for MRD detection [116]. When monitoring post-treatment ctDNA in the surveillance setting at multiple timepoints, both the sensitivity and specificity of CAPP-Seq for predicting relapse were 100% [116]. Of note, the detection of ctDNA MRD preceded radiographic relapse via routine CT imaging in 72% of patients by a median of 5.2 months [116]. These findings provide strong evidence for the potential of ctDNA MRD as a prognostic biomarker in patients with unresectable NSCLC after curative-intent treatment; moreover, they encourage future prospective studies to assess the clinical utility of ctDNA MRD in improving the selection of patients for consolidation therapy and possible treatment de-escalation.

In 2019, Chen et al. published findings from the DYNAMIC study (NCT02965391) [118]. This trial aimed to investigate perioperative ctDNA dynamic changes in patients with NSCLC after an R0 resection (i.e., microscopically margin-negative) [118]. The authors applied cSMART, which utilizes multiplex PCR using a panel of eight commonly mutated genes in NSCLC, to amplify plasma ctDNA from 205 study participants, of whom 36 had detectable ctDNA in their preoperative samples [118]. As expected, both plasma ctDNA concentrations and mutant allele fractions rapidly decreased after surgical resection, suggesting that post-operative ctDNA reflects real-time tumor burden [118]. Next, the authors sought to determine the timepoint at which detectable ctDNA after R0 resection had prognostic value [118]. Based on the prospective follow-up of 25 evaluable patients with multiple plasma samples collected after surgery (e.g., 1, 3, and 30 days post-operatively), the detection of ctDNA MRD three days after surgery was significantly associated with worse recurrence-free survival (RFS) and OS, relative to undetectable ctDNA MRD at this timepoint (median RFS of 278 vs. 637 days, and median OS of 434 vs. 720 days, respectively) [118]. Of note, the authors did not find an association between ctDNA MRD detection at one day after surgery and outcomes, suggesting that this timepoint is likely too early to detect microscopic residual disease due to a high rate of false-positives [118]. These combined findings suggest that detectable ctDNA MRD as early as three days after surgery is prognostic for disease relapse. This timepoint may be used for ctDNA MRD analysis in future clinical trials assessing adjuvant treatment escalation and possible de-escalation for such patients.

In 2020, Moding et al. sought to assess the potential of ctDNA MRD as a biomarker to stratify which patients would receive the most benefit from consolidation immunotherapy after CRT completion [117]. Using CAPP-Seq [40], the authors sequenced plasma cfDNA from 65 patients with unresectable stage IIB-IIIB NSCLC who were treated with CRT, of whom 28 also received consolidation ICI [117]. Pre-treatment ctDNA was detected in 78% of patients who did not receive consolidation ICI and in 75% of patients who received consolidation ICI [117]. Among 22 patients with serial ctDNA samples collected after starting consolidation ICI, 6 out of 7 patients (86%) with detectable ctDNA early on-consolidation ICI (i.e., median of 11 weeks into consolidation ICI) developed progressive disease, compared to only 2 out of 15 patients (13%) with undetectable ctDNA at this timepoint [117]. Furthermore, detectable ctDNA during early on-consolidation ICI conferred a significantly higher risk of disease progression than undetectable ctDNA (FFP of 0% vs. 87.5%, respectively, at 12 months after starting CRT) [117]. These findings suggest that ctDNA analysis after starting ICI can detect residual disease and is prognostic of disease recurrence. Among patients with undetectable ctDNA after CRT, the authors did not observe a statistically significant difference in FFP, regardless of whether consolidation ICI was received or not (*p* = 0.23) [117]. In contrast, patients with detectable ctDNA after CRT showed significantly better FFP when consolidation ICI was received (*p* = 0.04) [117]. These findings suggest that ctDNA MRD may help select those patients who are most likely to benefit from consolidation ICI and enable those without residual disease to avoid the potential toxicity of ICI therapy. Of note, the study was not powered to detect a small benefit from consolidation ICI among patients with undetectable ctDNA. Future prospective studies with large sample sizes are required to demonstrate the non-inferiority of withholding consolidation ICI from these patients. Lastly, the authors found that changes in ctDNA concentrations (i.e., ctDNA dynamics) further clarified which patients among those with detectable ctDNA after CRT responded to consolidation ICI. Patients with decreasing ctDNA levels after starting consolidation ICI achieved a longer FFP, compared to those with increasing ctDNA levels (median of 22 vs. 5 months, respectively) [117]. Indeed, patients with increasing ctDNA levels, despite starting consolidation ICI, did not have a significantly better FFP (*p* = 0.47) than patients with detectable ctDNA who did not receive consolidation ICI [117]. Thus, ctDNA monitoring during consolidation ICI may help to identify the subset of patients whose tumors will fail to respond to immunotherapy and who might benefit from either treatment escalation or a change in systemic therapy. However, the authors only performed an analysis of plasma ctDNA at a single early on-consolidation timepoint, which is clinically important to facilitate earlier intervention in the setting of treatment resistance but may precede the timepoint when some patients respond to immunotherapy. Future studies should investigate multiple timepoints during consolidation ICI to confirm these findings.

Updated findings from the TRACERx study were presented at the American Association for Cancer Research Annual Meeting in 2020 [115]. In their follow-up analysis, the authors applied PCM™ (ArcherDx, Inc., Boulder, CO, USA), a tumor-informed ctDNA assay targeting 196 patient-specific tumoral variants using a personalized multiplex PCR to sequence plasma ctDNA [115]. Study investigators analyzed samples from 78 patients with resectable stage I–III NSCLC [115]. Among the 45 patients that developed disease relapse during follow-up, 82% had a positive ctDNA MRD assay before or at the time of disease recurrence, showcasing this assay’s high sensitivity for detecting disease relapse. Furthermore, only one of the 23 patients without disease recurrence had a positive ctDNA MRD during follow-up, demonstrating the high specificity of PCM™ to detect MRD [115].

In 2020, Zviran et al. developed MRDetect (C2i Genomics, New York, NY, USA), which overcomes the constraint on sensitivity imposed by a low disease burden after surgery [119]. The authors showed that a tumor-informed analysis, combining thousands of SNVs and CNAs detected by whole-genome sequencing (WGS) of tumor tissue [47,48], can achieve highly sensitive ctDNA detection in the case of tumor fractions as low as 10^−5^, despite a low depth of coverage [119]. They applied MRDetect to plasma samples collected from 22 patients with stage I–III NSCLC after surgical resection [119]. Among 5 patients who relapsed, all had detectable ctDNA, while 12 of the 17 remaining patients who were relapse-free did not have detectable ctDNA, showcasing this assay’s high sensitivity and reasonable specificity to predict disease relapse [119]. Future studies with larger patient cohorts will be important to validate the promising sensitivity of this WGS-based assay for patients with NSCLC after surgical resection.

In 2021, Kurtz et al. developed a new technology, termed Phased Variant Enrichment and Detection Sequencing (PhasED-Seq) (Foresight Diagnostics Inc., Aurora, CO, USA), to improve the sensitivity of ctDNA MRD detection [120,123]. Assays that utilize molecular barcoding, such as CAPP-Seq, rely on duplex sequencing to improve the limit of detection, but duplex strand recovery is often suboptimal in the MRD setting [41,124]. PhasED-Seq instead focuses on the detection of multiple mutations within the same strand of DNA (i.e., phased variants), foregoing the need to recover duplex strands to increase the efficiency of genome recovery [123]. Using this novel strategy, the authors showed that PhasED-Seq outperforms duplex sequencing with CAPP-Seq, yielding a limit of detection below one part per million [123]. PhasED-Seq was then applied to 14 plasma samples, collected from five patients with localized NSCLC, for MRD detection [120]. PhasED-Seq detected ctDNA MRD in 10 of the 14 samples, while a SNV-based ctDNA method only detected MRD in 5 of the 14 samples. Furthermore, the authors compared both methods to detect MRD in three longitudinal samples from a patient with unresectable stage III NSCLC who underwent curative-intent treatment with CRT and consolidation ICI [120]. While the SNV-based method failed to detect ctDNA MRD in all of the samples, PhasED-Seq detected ctDNA MRD at each of these timepoints [120]. These findings suggest the potential of PhasED-Seq as an ultrasensitive method for ctDNA MRD detection in patients with NSCLC. Future studies are warranted to validate its sensitivity in larger patient cohorts.

Xia et al. conducted a large prospective cohort study, aiming to validate previous findings that perioperative ctDNA MRD detection is prognostic for disease relapse after surgical resection in patients with stages I-III NSCLC [121]. Nine hundred and fifty plasma samples from 330 evaluable patients enrolled on the LUNGCA-1 study (NCT03317080) were collected and analyzed using a tumor-informed assay. The majority of patients had a diagnosis of stage I NSCLC; the most common histological subtype was lung adenocarcinoma (LUAD) [121]. Plasma samples were collected before surgery, 3 days after surgery, and 1 month after surgery. Only 20.9% of patients had detectable ctDNA before surgery. Among those, ctDNA MRD detection at either 3 days or 1 month after surgery was associated with a higher rate of disease relapse (HR = 11.1, *p* < 0.001) [121]. Xia et al. further demonstrated that pre-operative ctDNA positivity was also associated with lower RFS (HR, 4.2; *p* < 0.001), although, interestingly, it was only prognostic among patients with LUAD and not lung squamous cell carcinoma (LUSC) [121]. Of note, ctDNA MRD detection outperformed all other clinicopathologic variables studied, including TNM stage, for predicting disease relapse. Adjusting for these other variables, adjuvant therapy improved RFS for patients with positive ctDNA MRD detection [121]. Interestingly, patients with negative ctDNA MRD showed worse RFS when receiving adjuvant chemotherapy compared to the ones who did not receive it (HR, 3.1; *p* < 0.001) [121]. These results reinforce the prognostic value of ctDNA detection after surgery for patients with early-stage NSCLC and showcase its potential to inform adjuvant therapy selection and possible treatment de-escalation.

Waldeck et al. aimed to assess the optimal timepoint of blood collection for ctDNA analysis among patients with stages IA-IIIB NSCLC who were undergoing surgical resection [122]. The authors prospectively enrolled 33 patients with resectable NSCLC in their study. Twenty-one patients were evaluable for longitudinal ctDNA analysis before surgery, during surgery, 1–2 weeks after surgery, and during clinical follow-up [122]. The authors showed that samples collected during surgery, taken immediately before tumor resection, had higher ctDNA levels compared to samples collected before surgery [122]. They hypothesized that physical manipulation of the tumor during surgery contributes to ctDNA shedding. They also demonstrated that ctDNA MRD detection at 1–2 weeks after surgery is prognostic of disease relapse after curative-intent resection [122]. While other studies have supported the prognostic value of ctDNA MRD detection as early as 3 days after surgery, due to the rapid clearance of ctDNA after tumor resection [118], the authors of this study suggest that a later timepoint for collection at the time of hospital discharge or at the first post-operative follow-up may be more practical and may have a lower risk of false-positives.

Plasma genotyping techniques have significantly evolved over the past few decades. However, despite the improvement in genomic coverage offered by newer technologies, all detection methods face similar technical and biological challenges, including low amounts of ctDNA in plasma, high levels of non-tumoral cfDNA arising from endothelial cells and PBMCs, CH, passenger mutations, and background noise (e.g., DNA oxidative damage, PCR errors, and sequencing artifacts) [23,55]. While bioinformatics pipeline refinement has overcome certain technical challenges, such as background noise [41], CH represents a major challenge when trying to implement ctDNA assays for MRD detection. One way to distinguish CH from tumoral variants is to perform parallel sequencing of PBMCs and plasma and exclude those variants present in PBMCs from the list of variants detected in plasma, as previously described [116,125]. While passenger mutations may not be therapeutically relevant, their detection has been reported to be important for ctDNA MRD detection [116,125]. In the context of treatment monitoring, these mutations also play an important role as they may signal the emergence of treatment resistance. Given the technical and biological limitations of ctDNA testing (Table 1), we recommend carefully interpreting and applying results outside of a research context. Further studies with larger patient populations are warranted to establish the clinical utility of ctDNA as a biomarker to guide clinical decisions for patients with NSCLC who are treated with curative-intent, along with its role in early cancer detection.

In summary, ctDNA MRD detection has been shown by multiple groups to be a prognostic biomarker in patients with stages I-III NSCLC, and it remains to be seen whether ctDNA MRD detection can be used to predict treatment response and personalize treatment based on significant differences in survival outcomes. Multiple ongoing studies are prospectively assessing treatment personalization based on ctDNA MRD detection and will assess the clinical utility of ctDNA MRD as a biomarker for directing the clinical management of patients with NSCLC (Table 2) [126,127,128,129].

## 6. ctDNA for Early Cancer Detection

A low-dose CT scan of the chest (LDCT chest) has been the current standard for lung cancer screening in the US since the National Lung Screening Trial (NSLT) [130]. Within this randomized trial, which compared annual LDCT chest for 3 years versus a single chest radiography among asymptomatic participants at high-risk for lung cancer, the LDCT chest was associated with a 20% relative reduction in mortality from lung cancer [130]. However, not all patients will benefit from an LDCT chest as current guidelines do not recommend this screening test for patients without a smoking history or those who have not smoked for many years [130]. Furthermore, LDCT chest has a 96% false-positive rate, thus leading to an excessive use of diagnostic testing for asymptomatic individuals [130]. Incorporating ctDNA analysis into screening may help to address these issues by improving the detection of lung cancer in a broader patient demographic and reducing the number of diagnostic procedures.

The majority of LDCT chest findings are small pulmonary nodules and ground-glass opacities, which may not be malignant or may shed very low concentrations of ctDNA [131]. Akin to the challenges of ctDNA detection in the MRD setting, two major challenges for ctDNA detection during screening are a low VAFs, due to a small tumor volume, and distinguishing tumor-derived mutations from CH [131]. However, as opposed to MRD detection, screening does not provide the option of tumor-informed variant calling, thus making it difficult to detect true biological mutations with a low VAF [131]. As shown by data from the TRACERx study [39], ctDNA for a stage T1b NSCLC tumor of 1 cm is present at an estimated VAF of 0.008% (95% CI: 0.002–0.03%), which is just beyond the detection limit of most liquid biopsy technologies. Therefore, the early detection of stage ≤ T1b tumors via a tumor-naïve approach presents a significant challenge regarding the effective implementation of ctDNA assays in lung cancer screening protocols [132].

Based on ctDNA detection via CAPP-Seq, Chabon et al. developed a machine-learning method, termed “Lung Cancer Likelihood in Plasma” (Lung-CLiP), to estimate the likelihood that a given blood sample contains cfDNA derived from lung cancer, discriminated from risk-matched controls [131]. In addition to tumor-naïve sequencing with a panel of 255 genes that are recurrently mutated in lung cancer, the authors leveraged the unique properties of NSCLC-derived mutations relative to CH mutations (e.g., shorter cfDNA fragment size and tobacco-smoking-associated mutational signatures) to reduce the rate of false-positive variant calling [131]. Lung-CLiP was trained on a discovery cohort of 104 patients with early-stage NSCLC and 56 risk-matched controls who underwent annual LDCT chest screening [131]. At a specificity of 98%, the algorithm yielded a sensitivity of 41% in patients with stage I disease, 54% in patients with stage II disease, and 67% in patients with stage III disease [131]. Compared to tumor-informed ctDNA analysis, Lung-CLiP achieves a similar level of performance without requiring tumor tissue genotyping [131]. Next, the authors prospectively validated Lung-CLiP in a cohort of 46 patients with early-stage NSCLC and 48 risk-matched controls, all of whom were enrolled at a different institution [131]. This independent validation demonstrated a statistically similar level of performance within each stage of disease, thus supporting the external validity of Lung-CLiP to predict the likelihood of lung cancer among similar types of patients [131]. Since most of the patients enrolled in this study were smokers and displayed incidentally diagnosed lung cancer, further work will be important to study the applicability of Lung-CLiP to never-smokers and patients who otherwise undergo LDCT chest screening.

Methylation-based ctDNA analysis presents an alternative strategy for lung cancer screening that largely avoids the issue of false-positives from CH mutations [49,50,133,134]. As part of the Circulating Cell-free Genome Atlas (CCGA) study, Liu et al. demonstrated a method of targeted cfDNA methylation analysis for multi-cancer early detection [135]. The authors applied bisulfite sequencing targeting a panel of >100,000 informative methylation regions, identified from the first sub-study of CCGA, to plasma cfDNA samples from 6689 participants (2482 with cancer across > 50 different cancer types and 4207 without cancer) [135]. Using a methylation-based classifier developed from a training set of the cohort, independent validation yielded a specificity of 99.3%, with a <1% rate of false-positives across all cancer types [135]. Sensitivity of stage I-III disease detection was 43.9% in all cancer types and increased to 67.3% in a pre-specified set of 12 high-signal cancer types, including lung cancer [135]. As expected, the sensitivity of detection improved with increasing disease stage [135]. For lung cancer, sensitivity ranged from ~25% for stage I disease to ~90% for stage IV disease [135]. Of note, localizing the tissue of origin (TOO) was predicted with 93% accuracy [135].

Recently, Klein et al. published their findings from the third and final sub-study of CCGA, a large clinical validation of the methylation-based classifier among 4077 independently enrolled participants (2823 with cancer and 1254 without cancer) [51]. Similar to results reported from the second sub-study by Liu et al., the specificity for detecting cancer was 99.5% [51]. For lung cancer, the sensitivity of detection across all stages was 74.8%, ranging from 21.9% for stage I disease to 95.2% for stage IV disease. [51]. TOO was predicted with 88.7% accuracy [51]. Altogether, these results suggest that methylation-based analysis of ctDNA for multi-cancer early detection may complement existing screening tests for individual cancers. However, given that CCGA is a case-control study, further work will be important for studying the utility of this methylation-based test for screening populations. The ongoing SUMMIT study (NCT03934866) is specifically for UK NHS patients who are at high risk of lung cancer, all of whom will have ctDNA analysis and screening LDCT, with follow-up over several years [136].

In an effort to increase the sensitivity of early lung cancer detection, compared to targeted sequencing of either mutations or methylation in ctDNA, Mathios et al. performed a genome-wide analysis of cfDNA fragmentation profiles [52]. Their method, termed the DNA evaluation of fragments for early interception (DELFI) [137], combines an analysis of genome-wide cfDNA fragmentomes (i.e., evaluating the size distribution and frequency of millions of cfDNA fragments) with clinical risk factors and CEA levels within a machine learning model to detect lung cancer (Table 1) [52]. The authors assessed blood samples from 365 participants who were enrolled in a prospective observational trial (LUCAS cohort), most of whom were symptomatic individuals at high risk of lung cancer [52]. Among these individuals, 129 had lung cancer diagnosed by tissue biopsy after blood collection, 87 had benign nodules, and 149 were not biopsied due to the low clinical and radiographic suspicion of cancer [52].

Median DELFI scores were similar among individuals with benign lesions (0.21) versus those without a biopsy (0.16) and were significantly higher among patients with lung cancer (ranging from 0.35 to 0.99 for stage I to stage IV, respectively) [52]. Among all patients in the LUCAS cohort, the DELFI approach detected lung cancer with an area under the curve (AUC) of 0.90 [52]. The AUC increased to 94% when considering only the majority of patients at higher risk for lung cancer (i.e., age 50–80 with a >20 pack per year smoking history) [52]. The predictive performance of DELFI was externally validated in an independent cohort of 385 participants without cancer and 46 participants with predominantly early-stage cancer, and yielded similar levels of sensitivity and specificity as in those observed by the model applied to the LUCAS cohort [52]. Furthermore, the incorporation of genome-wide cfDNA fragmentation data from the binding sites of ASCL1, a transcription factor differentially overexpressed in small cell lung cancer (SCLC), may even enable SCLC patients to be distinguished from NSCLC patients with an AUC of 0.98 [52]. Altogether, these findings suggest that ctDNA analysis via DELFI could potentially enhance lung cancer screening [52]. The authors propose a schema in which patients with positive DELFI pre-screening may proceed to LDCT chest analysis for further diagnostic workup [52]. DELFI-L101 (NCT04825834) is an ongoing prospective trial aiming to evaluate a DELFI-based assay among individuals eligible for lung cancer screening [138]. Further work will be important for assessing the utility of DELFI in combination with LDCT chest and other markers of early lung cancer detection.

In conclusion, ctDNA analysis via mutational, methylation, and/or fragmentation profiles offers promising potential for improving early lung cancer detection [53,54]. While positive ctDNA screening alone may not be adequate, its minimally invasive nature can provide a valuable pre-screening tool for a broader patient demographic and increase the uptake and adherence of screening with LDCT chest. Perhaps, future studies for cancer interception may utilize ctDNA technologies, such as DELFI and Grail Galleri™, for the emergence of pre-cancerous lesions to lessen cancer burdens in high-risk patients with a smoking history [51,52]. Therefore, future clinical studies centered on validating multi-omic prediction models that integrate liquid biopsy markers and LDCT will be important for optimizing lung cancer screening moving forward.

## 7. Conclusions

Liquid biopsies using ctDNA have the potential to enable the delivery of highly personalized treatment for patients with NSCLC. As an alternative to tumor tissue genotyping, ctDNA may be used as a minimally invasive method for tumor molecular profiling to identify actionable driver mutations that can guide treatment decision-making. In addition, ctDNA can be used to identify patients who are responding to therapy or, conversely, developing disease progression due to new genomic alterations. For patients who undergo curative-intent therapy, ctDNA can detect MRD and is a strong prognostic biomarker for disease relapse, but its clinical utility remains to be proven. Lastly, the multi-omic profiling of ctDNA shows promising evidence for early cancer detection and may be an important screening tool in addition to LDCT chest imaging. A major issue in the field is the lack of technological standardization across different ctDNA assays. The technologies applied for ctDNA analysis can impact assay performance since the sensitivity and specificity of an assay vary substantially, depending on the limit of detection and bioinformatics pipeline. Before selecting an assay, clinicians and researchers need to be familiar with the technological limitations of the selected product and when to use such tests. While ctDNA analysis is widely used in clinical practice for molecular profiling, the clinical utility of these assays for treatment monitoring, MRD detection, and early cancer detection remains to be proven. Therefore, ctDNA assays should not be used outside a research context for these applications in patients with NSCLC. We anticipate that liquid biopsies will be incorporated into the “TNM” staging system for lung cancer as an emerging metric for more precise clinical staging, evaluation of treatment response, and MRD detection in the future. Further studies with larger patient populations are warranted to establish the role of ctDNA as a biomarker that can guide clinical decisions for patients with NSCLC who are treated with curative-intent, as well as its role in early cancer detection.

## Figures and Tables

**Figure 1 ijms-23-09006-f001:**
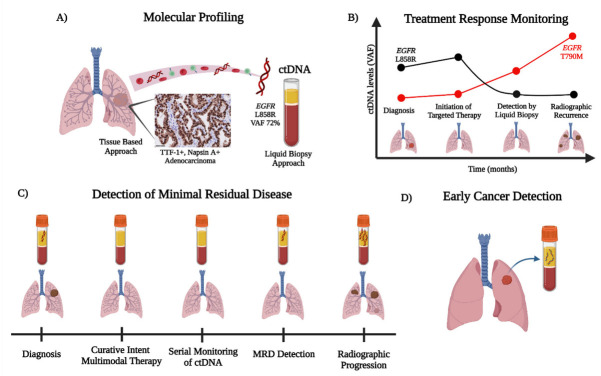
**Overview of ctDNA detection and the workflow for NSCLC**. The detection of ctDNA through liquid biopsies can be utilized for (**A**) molecular profiling, (**B**) treatment response monitoring, (**C**) detection of minimal residual disease, and (**D**) early cancer detection. (**A**) ctDNA can be readily used to identify cancer-related aberrations (e.g., detection of *EGFR* L858R) with ultrasensitive detection, for ease of use when tissue is limited or exhausted, or as a complementary diagnostic tool with rapid turn-around times. (**B**) The ease of ctDNA detection by serial blood draws permits treatment response monitoring of tumor biology evolution, via the detection of potential mechanisms of acquired resistance (e.g., *EGFR* T790M on first-generation EGFR TKIs). (**C**) ctDNA is a powerful tool for minimal residual disease (MRD) detection after curative-intent multimodal therapy (i.e., chemotherapy, radiation, and immunotherapy) in locally advanced NSCLC. ctDNA MRD positivity can be detected generally prior to radiographic progression to aid in decision making for patient care. (**D**) In an effort to improve early cancer detection in high-risk individuals, minimally invasive ctDNA analysis via mutational, methylation, and/or fragmentation profiles offers promising potential to complement radiographic screening with low-dose CT chest (LDCT chest).

**Figure 2 ijms-23-09006-f002:**
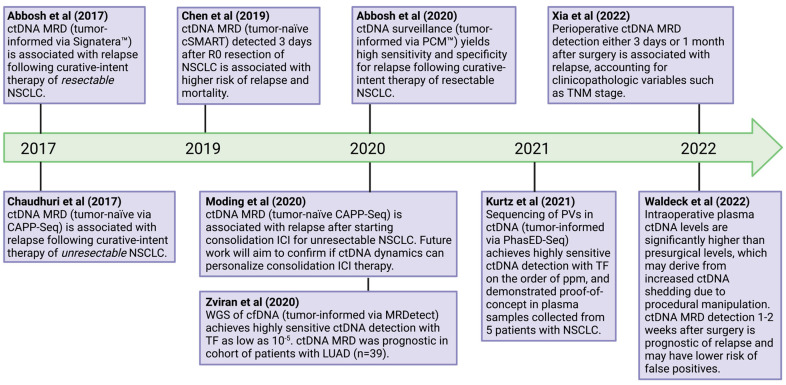
**Timeline of ctDNA MRD studies in patients with NSCLC.** A timeline of studies that have advanced our understanding of ctDNA MRD as a prognostic and potentially predictive biomarker in patients with NSCLC, who were treated with a curative-intent [39,115,116,117,118,119,120,121,122].

**Table 2 ijms-23-09006-t002:** **Ongoing** **clinical trials actively investigating the role of ctDNA in personalized treatment in patients with NSCLC.**

Trial (NCT #)	Primary Treatment	Stage	Estimated Enrollment	Recruitment Status	ctDNA Timepoint	ctDNA (+) Intervention	ctDNA (−) Intervention	Phase	Primary Endpoint	Type of Assay
SCION (NCT04944173)	SBRT+ 4 cycles of durvalumab	T1-2 N0 M0	94	Not yet recruiting	After SBRT + 4 cycles durvalumab	Additional 8 cycles of durvalumab	No further treatment	II	ORR at 18 months	Avenio
NCT04585490	CRT	III	48	Recruiting	After CRT	Four cycles of platinum doublet chemo + durvalumab (1500 mg IV every 21 days, for 1 year)	SoC durvalumab (10 mg/kg every 2 weeks, or equivalent, for 1 year)	III	Change in ctDNA level following chemo	Avenio
NCT04585477	Surgery or definitive SBRT	I–III	80	Recruiting	After surgery or SBRT	Twelve cycles of durvalumab	SoC and no treatment	II	Decrease in ctDNA level	Avenio
MERMAID-1 (NCT04385368)	Resection of primary NSCLC	II–III	332	Active, not recruiting	After surgery	Durvalumab + SoC chemo vs. placebo + SoC chemo	N/A	III	DFS in MRD+ analysis set	ArcherDx
MERMAID-2 (NCT04642469)	Surgery +/− neoadjuvant or adjuvant Tx	II–III	284	Active, not recruiting	After surgery	Durvalumab vs. placebo	N/A	III	DFS in the PD-L1 TC ≥ 1% analysis set	ArcherDx

Abbreviations: chemo, chemotherapy; DFS, disease-free survival; MRD, minimal residual disease; NSCLC, non-small cell lung cancer; ORR, overall response rate; SBRT, stereotactic body therapy; Tx, treatment.

## Data Availability

Not applicable.

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
