# Peer review of "Making the Rounds: Exploring the Role of Circulating Tumor DNA (ctDNA) in Non-Small Cell Lung Cancer"

_ijms, 2022, doi:10.3390/ijms23169006_

Round 1

Reviewer 1 Report

In this review, the authors focused their attention on liquid biopsy and its potential clinical applications in NSCLC patients, including cancer diagnosis, treatment and minimal residual disease detection. Most studies actually have indicated that ctDNA testing is very important in diagnosing NSCLC, predicting clinical outcomes, and response to targeted therapies and immunotherapies, and detecting cancer recurrence. It is very interesting the discussion about the potential role of liquid biopsy in improving early lung cancer detection taking into account the importance of DNA analysis via mutational, methylation, and/or fragmentation profiles. Shields and colleagues resumed the current applications of ctDNA for the management of NSCLC and discussed future directions for guiding clinicians in better treatment decision-making.

To my eyes, the paper is good by considering the large number of studies proposed, ongoing clinical trials that actively investigate the role of ctDNA and personalized treatment in patients with NSCLC. The scientific content seems good as well as the English style and language used in the manuscript need. Moreover, I appreciate the scientific efforts to organize this paper and I think that the rationale is well-discussed. The resolution of the table of each section and the reference list meets the quality requirements of our journal. Moreover, the results were clearly discussed and corroborated with what is shown. Finally, the data analyses were interpreted in a comprehensible manner. I didn’t observe any remarkable incongruences throughout the text.

In my opinion, this article should be considered with minor revisions in light of comments/suggestions as indicated below.

1-      Since the liquid biopsy is a rapidly evolving diagnostic tool for precision oncology that has recently found its way into routine practice as an adjunct to tissue biopsy, I think that it could be important to consider pragmatic scenarios for the use of ctDNA from blood plasma to identify actionable targets for therapy selection in NSCLC patients. It is well established that different clinical scenarios may require different analysis strategies but key preanalytical considerations for ctDNA mutation testing in NSCLC could probably impact the performance. I would ask the authors to integrate the preanalytical steps of ctDNA testing and explain how different many factors can affect it.

2-      It’s ascertained that the choice of method is also very much dependent on the specific application, consensus is needed on which methodologies may provide the best answers at various stages of the patient journey. Are there any indications between intra- and inter-laboratory comparisons in order to minimize the variability in experimental procedure? Please fix this point.

3-      An important step in the delivery of precision oncology to patients with lung cancer is the interpretation and reporting of variants in the clinical context. There are various levels of complicating factors, starting with distinguishing between variants that might not be disease-related as well as PCR/sequencing artifacts, and passenger mutations, and proceeding to interpret low variant allele frequencies variants. Please deepen this topic.

4-      Such studies involving research into multianalyte blood tests for various cancer types are conceptually and practically being explored with promising results regarding ultrasensitive testing (e.g. CancerSEEK) to enhance early detection by liquid biopsy screening. What’s your consideration about it? And how could increase the clinical sensitivity in the future?

5-      One of liquid biopsy testing problem is linked to ctDNA mutant allele fractions that can be extremely low, so mutation-based detection of ctDNA has clear limitations for minimal residual disease and monitoring within the plasma. What are clinical strategies to overcome this limitation? Please reply to it in the text.

6-      Please recheck the reference list, fix some abbreviations and punctuation throughout the text.

Author Response

We thank the reviewers for their constructive feedback, which has helped us significantly strengthen the manuscript. Please find attached our point-by-point responses to the reviewers' comments (highlighted in blue text).

Reviewer 2 Report

The manuscript entitled:" Making the Rounds: Exploring the Role of Circulating Tumor 2 DNA (ctDNA) in Non-Small Cell Lung Cancer" fcused o na review of literature data about the main field where ctDNA is currently applied focus on a timely relevant topic but moderate integrations shoudl be performed to publish this manuscript on this journal

- In the introduction section, please, could the authors provide mroe details about the other conventional analytes traced in peripheral blood? In addition, could they improve this section with some biological impelementations related to the orgins of ctDNA? Accordingly, the authors should elucidate the main limitations and point of strenght that derive from the analysis of this analyte in clinical practice.

- In the text, i would reccomend to report a dedicated section for the main technical approaches able to analyze ctDNA i neach clinical section. I would reccomend to improve this section in order to focus on the technical issues found in each clinical setting for ctDNA analysis.

- In the text, the authors (lines 197 -118) the authors show a technical approach NGS ultra deep based built to evaluate very low frequency mutation harboured by ctDNA. In my opinion, the authors should focus on solid NGS assay that could met clinical need in this setting because this approach seems not adequate to cover diagnostic requirements i nterms of suitability for ctDNA prediction analysis.

- In the future perspective section, please could the authors debaet about cancer interception and blood TNM as emerging aspects that coudl udnerline the role of blood in the managment of lung cancer patients?

- Please, could the authros improve the article with an imagine that schematize all critical aspects analzyed by the review?

Author Response

(The authors gave the same response as above.)

Round 2

Reviewer 2 Report

The manuscript is now suitable in the present form